# Dielectric and Impedance Studies of (Ba,Ca)TiO_3_ Ceramics Obtained from Mechanically Synthesized Powders

**DOI:** 10.3390/ma12244036

**Published:** 2019-12-04

**Authors:** Kamil Feliksik, Lucjan Kozielski, Izabela Szafraniak-Wiza, Tomasz Goryczka, Małgorzata Adamczyk-Habrajska

**Affiliations:** 1Faculty of Science and Technology, Institute of Materials Engineering, University of Silesia, 75 Pułku Piechoty St. 1A, 41-500 Chorzów, Poland; kamil.feliksik@us.edu.pl (K.F.); tomasz.goryczka@us.edu.pl (T.G.); malgorzata.adamczyk-habrajska@us.edu.pl (M.A.-H.); 2Institute of Materials Science and Engineering, Poznan University of Technology, Jana Pawła II 24, 61-138 Poznań, Poland; izabela.szafraniak-wiza@put.poznan.pl

**Keywords:** mechanochemical synthesis, lead-free piezoelectric material, BCT ceramics

## Abstract

Mechanochemical synthesis offers unique possibility of perovskite phase formation at ambient conditions that is very attractive (simplifies production, allows strict stoichiometry control and brings economic benefits). In this work the mechanochemical synthesis has been used for preparation ofBa_1−x_Ca_x_TiO_3_ (0.2 ≤ x ≤ 0.3) powders from simple oxides. The 20 h milled powders have been uniaxially pressed and sintered in order to get the ceramic samples. The sample morphologies have been observed by scanning electron microscopy. Dielectric and impedance studies have been performed on ceramics. The obtained results indicate that the two mechanism of doping occurred. The first one is observed for the lower calcium concentration (below 0.3) and consists of the introduction of calcium ion into the A site of the perovskite structure. The second one is observed for the higher calcium concentration (equal 0.3). In this case the calcium ions partially occupies the B site in the perovskite structure. Both cases have different influence on the final properties of the ceramics because they induce different defects.

## 1. Introduction

Currently the most of daily used devices contains electroceramic parts. From the industrial point of view the lowering of process temperature, both during preparation and densification stages, and the shortening of high temperature treatments are desirable. One of the most promising routes is based on mechanochemical synthesis (MS). The main advantage of this method is relatively low temperature of processing, which can be done in the room temperature. The additional benefit of MS approach is the usage of simple oxides. They are cheaper and reveal better chemical stability than compounds needed for wet chemical methods [1,2].

In conventional high temperature solid state synthesis, used in ceramic material technology, reactive molecules encounter one another through random thermal motion during the thermal diffusion process. In the MS process the reduction of the grain sizes of the starting oxides results in the formation of high densities of defect, shorter diffusion distances, nearer contacts of grains and creation of fresh/cleaned surfaces/interfaces. In consequence, the enhancement of powder reactivity and the reactions of desired phases occur [3,4,5]. Hence, the usage of mechanochemical synthesis prevents unwanted reactions by keeping potential reactants at lower temperature prior to their evaporation. Consequently, this way of material production enables strict control of stoichiometry, because undesirable volatile elements losses are eliminated [5].

The widely investigated piezoelectric/ferroelectric materials are lead free BaTiO_3_-based perovskites. To improve their useful properties different doping or solid solutions have been analyzed. One of the most effective material for energy transformation and energy harvesting applications is barium calcium titanate (Ba,Ca)TiO_3_ (BCT) solid solution (especially with 20 at.% content of calcium). The BCT piezoelectric constant reaches the value of d_33_ = 620 pC/N, which is higher than PZT [6]. The surprisingly high piezoelectric coefficients of BCT are observed in the vicinity of the morphotropic phase boundary (MBP). In this composition range the polarization anisotropy nearly vanishes and polarization rotation occurs between <001> tetragonal and <111> rhombohedral states [7,8,9].

The pure BaTiO_3_ and its solid solution ceramics have been obtained by mechanochemical synthesis [9,10]. Recently, our work has been focused on mechanochemical synthesis of BCT ceramics and we have investigated the influence of mechanochemical condition on the phase formation [11]. The present work has been focused on final properties of the Ba_x_Ca_1−x_TiO_3_ (BCT, x = 0.2, 0.25 and 0.3) ceramics obtained from mechanochemically synthesized powders. We have performed detailed dielectric and impedance studies. Based on the obtained results, the mechanism of substitution of calcium ions in the BaTiO_3_ structure has been discussed.

## 2. Materials and Methods 

Various compositions of BCT were chosen for investigations: Ba_0.8_Ca_0.2_TiO_3_—denoted as BCT20, Ba_0.75_Ca_0.25_TiO_3_—BCT25 and Ba_0.7_Ca_0.3_TiO_3_—BCT30. 

The high-purity oxides BaO (99.9% pure, Sigma-Aldrich), CaO (99% pure, Avantor Performance Materials, Gliwice, Poland) and TiO_2_-anatase (99% pure, Sigma-Aldrich, Darmstadt, Germany) were weighed out in the desired molar ratios. The constituent oxides were of standard powder particle average sizes in the range of 10–50 microns. All the powder samples were subsequently milled in the high-energy ball mill SPEX 8000 (SPEX® SamplePrep, Metuchen, NJ United States) for 20 h (according to [11]). Linear shrinkage rates facilitate the study of sintering kinetics involved in the calcium and barium titanate thickening process. Sintering studies (Ba, Ca) TiO_3_ were performed in a NETZSCH DIL 402 C dilatometer (NETZSCH Analysing & Testing, Selb, Germany). Linear shrinkages for the green BCT samples after dilatometric experiments is clearly related to the compositions of investigated BCT. 

Thus, increasing the amount of calcium in investigated materials results in a slight decrease in the initial compaction temperature, an increase in temperature where the maximum densification rate occurs, and marked reduction in overall shrinkage. Total shrinkage of samples, temperature of densification start T_sd_ and temperature of maximum densification T_md_ for individual samples are summarized in Table 1, neither hardness nor fracture toughness were examined. A BCT20 sample with a near theoretical density (over 98%) was obtained. The BCT30 sample is more porous than others, probably due to the coexistence of two phases of barium titanate and calcium titanate with markedly different grain size as well as chemical and physical properties.

The presence of perovskite phase has been confirmed by X-ray powder diffraction (X’Pert Philips-Pro PW3040/60, PANalytical B.V., Almelo, Netherlands). The BCT powders were formed into pellets (6 mm in diameter and 2 mm thick) and subsequently compressed at 300 GPa using a cold isostatic press. Finally, the pressed pellets were sintered at 1400 °C for 3 h (HTC 1500 Ströhlein Instruments oven, Kaarst, Germany) without any pre-calcinations step. The heating rate was 5 °C min^−1^, with natural cooling in air atmosphere.

The obtained morphologies of the powders were examined using a scanning electron microscope with an ultra-high resolution JEOL JSM-7100F TTL LV (JEOL, Ltd. Tokyo, Japan) with a cold field emission source. SEM powders were prepared by dissolving in ethanol in an ultrasonic bath. Distribution of all the elements throughout the grains was examined by energy dispersive X-ray spectrometer (EDS, JEOL, Ltd. Tokyo, Japan). X-ray diffraction patterns of the ceramic samples were measured using diffractometer X’Pert Pro with copper radiation. The diffractograms were recorded in a step-scan mode (step 0.05 degree) at 2θ range from 15 to 135 degrees. The duration of signal collecting at the step was adjusted to receive proper statistic of the counts. The XRD measurements were done at room temperature.

The dielectric and impedance spectroscopy (IS) measurements were done on the polished samples of 0.6 mm thickness, coated by the gold electrodes, which were deposited on them by cathode sputtering. Before the measurements all samples were de-aged by thermal treatment at 450 °C to allow the recombination and the relaxation of the frozen defects, formed during the sintering process. The dielectric data as well as impedance once were obtained in the 20–2 × 10^6^ Hz frequency range using an HP 4192A (Keysight Technologies, Santa Rosa, California, United States) impedance analyzer.

## 3. Results and Discussion

### 3.1. SEM and EDS

The SEM microstructure of the mechanochemically synthesized powder of BCT25 and BCT30 are presented on Figure 1. The powders after mechanochemical synthesis consist of irregular, nanoscale particles with wide distribution (from 30 to 200 nm). The edges of individual particles were not sharp and they were often rounded. The strong van der Waals attraction between a large number of the particles led to polynuclear growth. The single particles exhibited a cubic-like morphology and showed a strong tendency to form agglomerates.

The ceramic process consisting of uniaxially pressing and sintering of the powders modifies the final morphology of samples (shown on Figure 2). The ceramics consisted of well-shaped grains with the visible terraces, which indicated the tendency to spiral hexagonal growth. This two-dimensional mechanism of grain growth results in significant increase of single grains. In consequence, increases in the strength of the obtained ceramics were noticed. Compared to the ceramics, obtained by other authors, ceramics consisted of the smaller grains with various shape [12,13], the mechanochemically based ceramics exhibit different morphology. Moreover the SEM studies of BCT20 ceramics reveal homogenous distribution of grain size. The increase of calcium content affects the final morphology, which exhibits the significant increase of grain heterogeneity. The average grain size was varied (Figure 2c) significant—from about 20 to 1 µm. The compositional analysis was performed using energy-dispersive X-ray spectroscopy EDS (Figure 2d). During this study the contents of barium, calcium and titanium were analyzed. The oxygen was omitted, because oxygen excess is present in pores (originate from the atmosphere) and interferes with the results. The content of investigated elements was normalized to 100%. The scans were done on 50 points of several grains. The example of EDS analysis, obtained for one of the point of grain for BCT20 ceramics has been shown in Figure 2d. The average results of quantitative analysis were collected in Table 2.

### 3.2. XRD Measurements and Dielectric Spectroscopy

The Ca substitution in barium titanate can lead to the formation of P4mm and R3c perovskite phases. The detailed analysis of the XRD diffraction patterns was performed for all ceramic samples (Figure 3). For lower content of the Ca (up to 0.25) the ceramics exhibited the single phase. For the Ca concentration equal to 0.3 two phases, tetragonal and orthorhombic ones could be identified in the materials. In order to explain the effect of Ca content on the crystallographic structure of ceramics, calculations of structural parameters were made using the Rietveld method. Obtained results were collected in Table 3. The crystal structure could be described using the tetragonal lattice with the P4mm space group and the lattice parameters: a = b = 3.9827(9) Å and c = 4.025(1) Å and a = b= 3.9756(9) Å and c = 4.020(1) Å, for BCT20 and BCT25 samples, respectively. For the ceramic with higher Ca content of up to 0.3 the tetragonal lattice reveals following parameters: a = b = 3.977(1) Å and c = 4.024(2) Å. The crystal lattice for the second phase was determined as the orthorhombic one with the R3c space group and lattice parameters: a = b = 5.73(3) Å and c = 14.22(9) Å. Due to the difference in the ionic radii of Ca^2+^ = 114 pm and Ba^2+^ = 149 pm the tetragonality (c/a) parameter had increased and its value changing from 1.0106 (x = 0.2) to 1.0118 (x = 0.3). Consequently also volume of the unit cell decreased for samples as the effect of Ca substitution in barium titanate (Table 3).

The results have proved that the average stoichiometry, measured for all samples, was consistent with nominal content of elements within the error limits, which for this method was equal ~1%. The more detailed analysis of obtained data revealed that the atomic content ratios of (Ba + Ca)/Ti in case of BCT20 was equal to 0.96, whereas for BCT30 was higher than 1. It indicates the change of the Ca^2+^ substitution way. Consequently the part of calcium ions replaced Ti^4+^ instead Ba^2+^ [12]. This difference can be related to the changes of phase transition character. In this situation the diffusive phase transition was present instead of the classical sharp one. This phase transition character change was usually clearly visible in dielectric studies. The temperature dependencies of the real part of dielectric permittivity (ε′) and loss tangent (tanδ) were investigated to confirm this assumption.

The typical curves of ε′(T) and tanδ(T) for Ba_1-x_Ca_x_Ti_O3_ ceramics with x = 0.2; 0.25 and 0.30 are shown in Figure 4. All samples show the maximum of dielectric permittivity (ε′max), which was correlated to ferroelectric-paraelectric phase transition (Figure 4a). The T_C_ (the temperature when ε′max was observed) and maximum values of dielectric permittivity ε′max were collected in Table 4. The presented data indicates the marginally influence of calcium ions on discussed values, especially the ε′max value remained almost unchanged for BCT20 and BCT25 ceramics. The fact has already been noted by other authors, who suggested the lack of Curie point changes related to the Ca concentration if x is less than 0.5 [14]. The fact that the Curie temperature doesn’t changeconfirms the corrected and expected incorporation of calcium ions in the A site of the perovskite structure. In case of the substitution Ca^2+^ ions in Ti^4+^ site in BaTiO_3_ lattice the drastic decrease in the dielectric maximum temperature should be observed [15,16]. The visible increase of the maximum value of dielectric permittivity for BCT30 with a simultaneous slight change in temperature TC indicates that a small part of the Ti^4+^ ions was substituted by Ca^2+^ ones. The fact was consistent with the EDS analysis results. 

Above the Curie temperature the 1/ε′ (T) dependence for classic ferroelectric follows the Curie–Weiss law:(1)$$\varepsilon =\frac{C}{T+T_{o}},$$
where T_0_ is the Curie–Weiss temperature and C is the Curie–Weiss constant. The temperature inverse ε′ plots for all investigated ceramics were fitted to the mentioned law. The curves presented on Figure 5 fulfilled well the law starting from the temperature T_dev_ significantly higher than T_C_ (Table 4). The obtained values of T_0_ and C parameters as well as T_dev_ were collected in Table 4. 

The large differences between T_C_ and T_de_v are associated with the diffuse character of phase transition. However, in the case of the investigated samples the T_0_ remained lower than T_C_, which suggests the small degree of broadened and lack of characteristic properties for the ferroelectric relaxor.

In the narrow temperature range (ΔT = T_dev_ − T_C_, Table 4), where the diffuse character of the ferroelectric–paraelectric phase transition caused the deviation from typical Curie–Weiss law the dependence 1/ε(T) could be described by modifying the Curie–Weiss relationship:(2)$$\frac{1}{\varepsilon }-\frac{1}{\varepsilon_{\max}}=\frac{{\left(T-T_{C}\right)}^{\gamma }}{C_{1}}$$
where γ is a diffuse parameter, which gives information on phase transition character. For γ = 1 the phase transition has a classical character, whereas the γ = 2 described a complete diffuse phase transition [17]. Figure 6 shows the plot of ln(1/ε′ − 1/ε_max_) versus ln(T − TC) for BCT20 and BCT30, the red lines are the curves obtained on the base on Equation (2).

The value of the diffuse parameter (see Table 4), obtained by fitting the experimental data based on Equation (2), had suggested the gradual increase of diffuseness with an increase of calcium ions, which could be related to the incorrect substitution of Ca^2+^ ions in B site of perovskite structure. Consequently this ion incorporation promotes the existence of the local compositional fluctuations that change locally the phase transition temperature and broaden the phase transition.

The substitution of Ca^2+^ with Ba^2+^ was homovalent and did not create a charge defects in perovskite structure, whereas Ca^2+^ substituted for Ti^4+^ in the BaTiO_3_ lattice and it acted as an acceptor to trap the unlocalized electrons. It created a doubly charged acceptor center Ca″_Ti_, which compensates the oxygen vacancies formed during sintering [18,19]:

$$BaO+CaO\to Ba_{Ba}+Ca_{Ti}^{\prime \prime }+2O_{o}+V_{o}$$

The proposed scenario should have impact on the electric properties of ceramics (especially grains and grain boundaries). The measurements of impedance spectroscopy in wide temperature range have been carried out in order to determine the temperature dependence of the resistance of grains and grain boundaries. The obtained impedance (*Z**) data are presented on Figure 7.

The shape of frequency dependencies of the real and imaginary part of complex impedance did not change significantly with the increasing of calcium content. The strong temperature dependence of Z_re_(f) was observed for all measuring range of frequency (Figure 7a,c,e). The Z_re_ values decrease with increasing temperature due to the conducting loss increase [20]. The slope of the Z_re_(f) dependencies was not very distinct in low frequency range. The Z_re_ values decreased rapidly at the middle frequency range. The Z_im_(f) dependencies of BCT20 had two local maxima that were temperature dependent—both maxima shift to high frequency with temperature increasing. The increase of the calcium content caused a gradual decline of the first maximum at a low range of temperature.

The further impedance data analyses were done based on the Nyquist plot. The Nyquist plots of all investigated ceramics at several temperatures are presented on Figure 8. The plots consisted of two well separated semicircles. Such a shape of Nyquist dependences suggests the presence of two different conduction mechanisms in the measured frequency and temperature range [21,22,23]. The semicircle in the high frequency region was related to the electric properties of the grains, the second one (in the low frequency region) was related to the grain boundary. The Z_im_(Z_re_) dependencies could be fitted to the equivalent circuit with two branches in the series that are related to the electrical properties of grain and grain boundary. For the ideal “Debye”-like ceramics the discussed branches consist of resistor and capacitor. In the real cases the ceramic materials do not manifest the ideal “Debye” like behavior. For better description of the realistic materials (that are deviated from an ideal capacitor model) the additional element with the constant phase (CPE) should be introduced into equivalent circuit (Figure 8d) [23].

The fitting of the impedance spectroscopy data gives the values of grains (R_G_) and grain boundary (R_GB_) at different temperatures. The example of such fitting is given in the Figure 9. The obtained values of R_G_ and R_GB_ decrease with the increasing temperature, which is associated to increases in conductivity with temperature. Moreover for BCT20 and BCT25 ceramics R_GB_ is higher than R_G_, which may be due to lower concentration of oxygen vacancies and trapped electrons in the grain boundaries [24]. The situation drastically changes for BCT30 ceramics, where the R_GB_ are lower than R_G_ (even two times lower at high temperatures).

The raw and data fitting were in fairly good agreement with what confirms the appropriate choice of equivalent circuit. The linear character of lnR_G_(1/T) and lnR_GB_(1/T) dependencies was related to the activation character of processes occurring in the grain and grain boundary (Figure 10). The slope of dependencies stayed almost unchanged for BCT20 and BCT25 ceramics over the entire investigated temperature range, which indicates that the activation energy of grains and grain boundary stayed at the same level. As a consequence in those ceramics at higher temperature the current flowed both by grain boundaries and grain interiors in the same level. The situation changed for BCT30 ceramics. In the ceramics two different mechanisms of the electric conductivity were observed—one at the high and second at the low temperature range. Moreover the significant decrease of the grain boundary activation energy indicates the increase in the number of defects agrees with the partially incorporation of the calcium ions in the place of the barium ones. 

## 4. Conclusions

Polycrystalline Ba_1−x_Ca_x_TiO_3_ (BCT, x = 0.2, 0.25 and 0.3) ceramics were successfully obtained via mechanochemical synthesis. The microstructure of the resulting ceramics consisted of very well developed grains with visible terraces, which indicates a tendency to spiral hexagonal growth. The increase of calcium content affected the final morphology, which exhibited a significant increase of grain heterogeneity. 

The results obtained suggest a change in the calcium substitution mechanism for BCT30 ceramics. As the concentration of calcium was lower than x = 0.3 all calcium ions substituted barium ones in the A site of perovskite structure. For x = 0.3 the calcium ions substituted both barium ones in the A and titanium ones in the B site of perovskite structure. The substitution mechanism correlated to the temperature changes of dielectric permittivity and impedance analyses recorded in a wide range of temperatures and frequencies.

However, for the purpose of this experiment, the conventional mixed oxide ceramic (MOM) method was used as the first approach despite the fact that, for example, rapid sintering can significantly reduce processing time. As shown above, high temperatures and long reaction times in the MOM method gave ceramics with very heterogeneous particle sizes. With the miniaturization of electronic devices, there is a great industrial need to control size effects, especially when approaching the nanosize scale. According to a report on a similar solid solution of Ba_1-x_Sr_x_TiO_3_, the combination of mechanosynthesis and SPS offers compaction that took place in a few minutes, demonstrating that synergy occurs due to the combination of the SPS process and mechanosynthesized powders, which is why this method is planned in the future [25].

## Figures and Tables

**Figure 1 materials-12-04036-f001:**
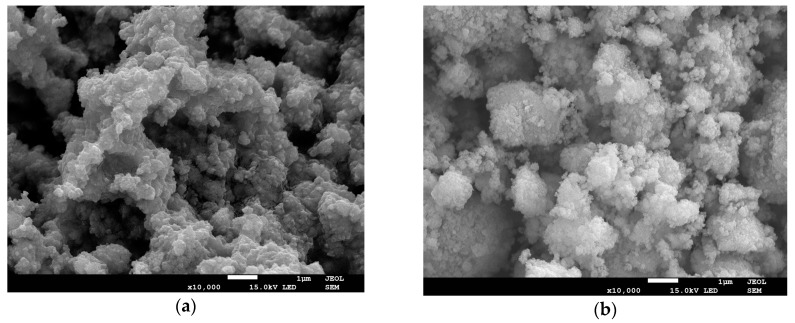
SEM micrographs of the BCT25 and BCT30 powders after 20 h of high energy milling: (**a**) BCT25 ceramic nano powder and (**b**) BCT30 ceramic nano powder.

**Figure 2 materials-12-04036-f002:**
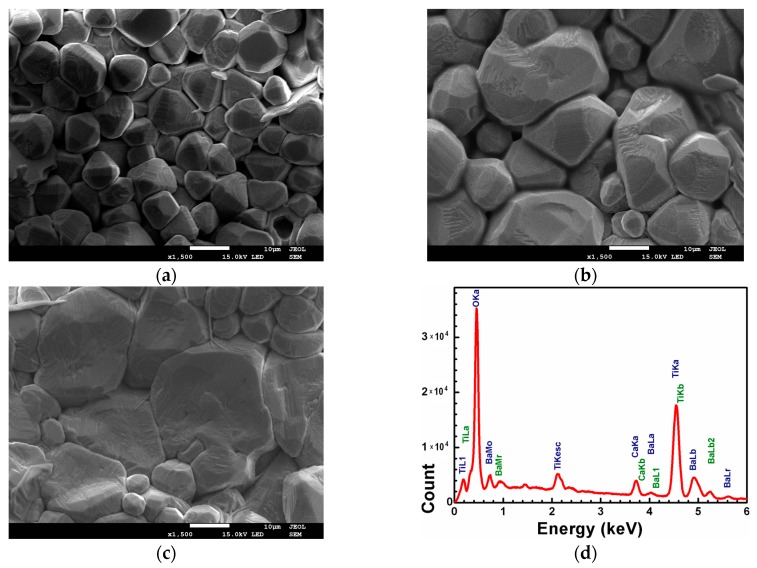
SEM micrograph of the BCT20 (**a**), BCT25 (**b**) and BCT30 (**c**) ceramics. The energy dispersive X-ray spectrometer (EDS) analysis obtained for one of the point of BCT20 ceramics grains (**d**).

**Figure 3 materials-12-04036-f003:**
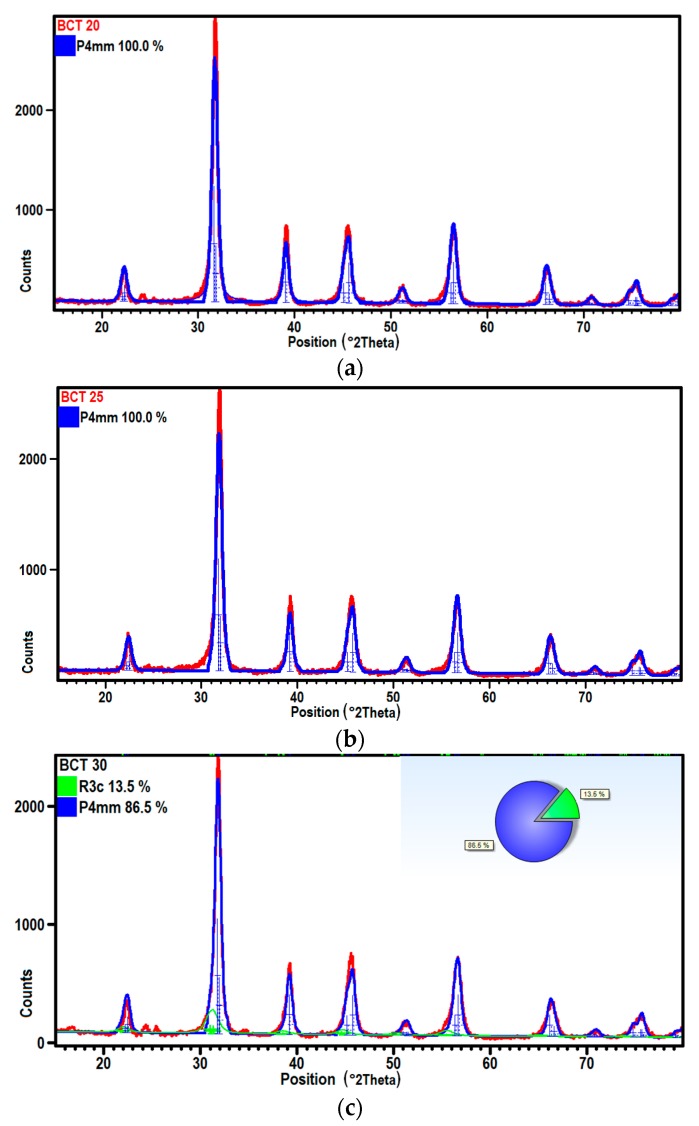
Result of the Rietveld refinement of XRD diffraction patterns done for BCT20 (**a**), BCT25 (**b**) and BCT30 (**c**) ceramics (red—experimental data, blue and green—calculated data).

**Figure 4 materials-12-04036-f004:**
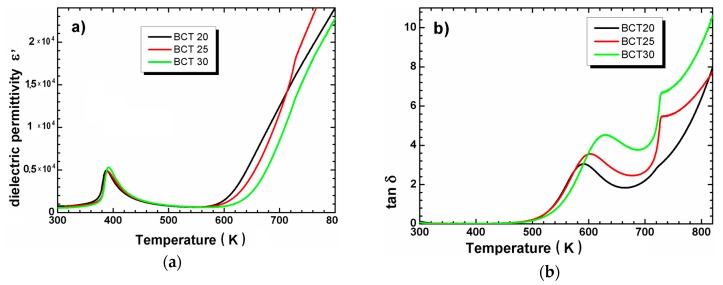
Temperature dependencies of dielectric permittivity (**a**) and loss tangent (**b**) measured during heating at frequency 10 kHz for BCT20, BCT25 and BCT30 ceramics.

**Figure 5 materials-12-04036-f005:**
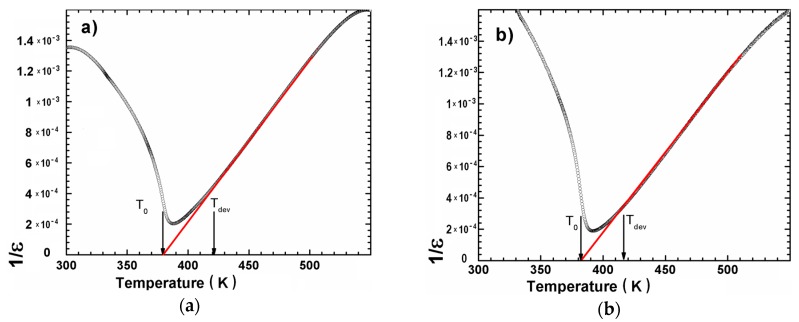
Dependencies of inverse of dielectric permittivity (1/ε) for the BCT20 (**a**) and BCT30 (**b**) ceramics (symbols—experimental data; solid line—fitting to the Curie–Weiss law.

**Figure 6 materials-12-04036-f006:**
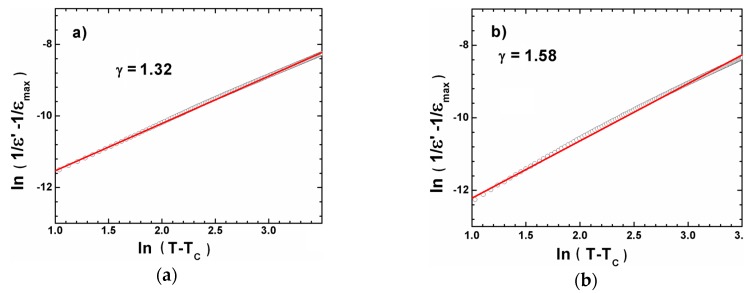
Dependencies of ln(1/ε′ − 1/ε_max_) as a function of ln(T − T_C_) for BCT20 (**a**) and BCT30 (**b**) ceramics. The solid line is the fit using Equation (2).

**Figure 7 materials-12-04036-f007:**
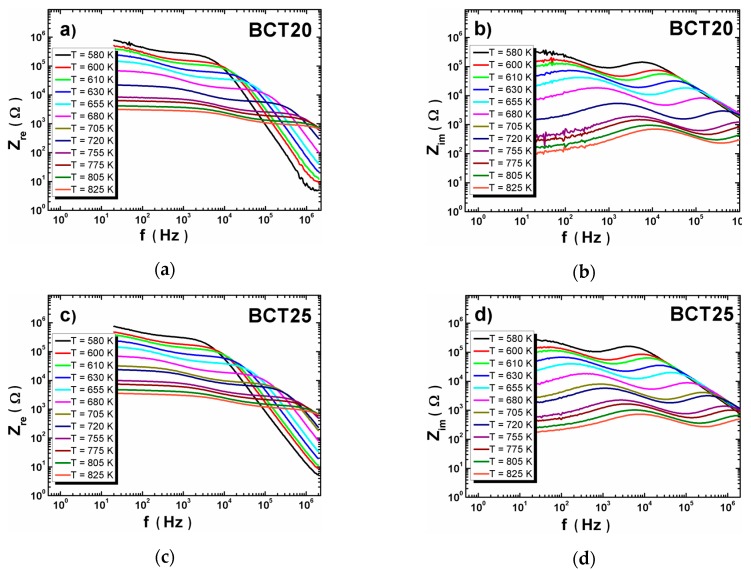
Dependence of the imaginary (**a**,**c**,**e**) and real (**b**,**d**,**f**) part of the impedance at various temperatures for BCT20, BCT25 and BCT30 ceramics.

**Figure 8 materials-12-04036-f008:**
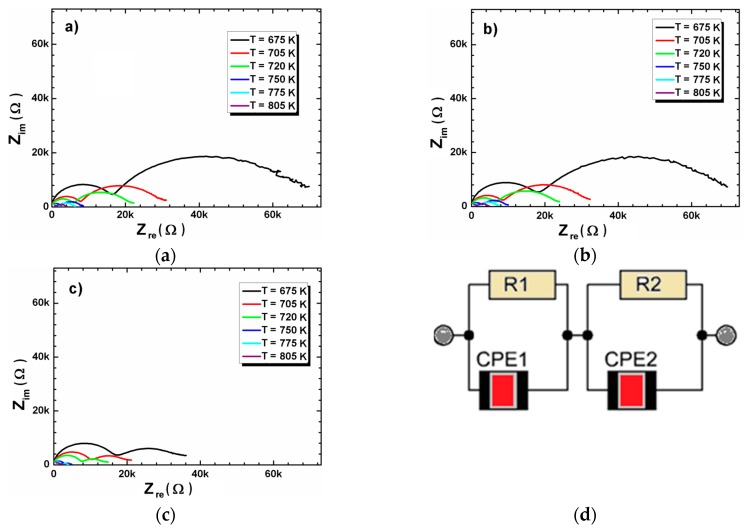
Nyquist plot for BCT20 (**a**), BCT25 (**b**) and BCT30 (**c**) ceramics presented at several temperatures. Equivalent circuit used to represent the impedance response of BCT ceramics (**d**).

**Figure 9 materials-12-04036-f009:**
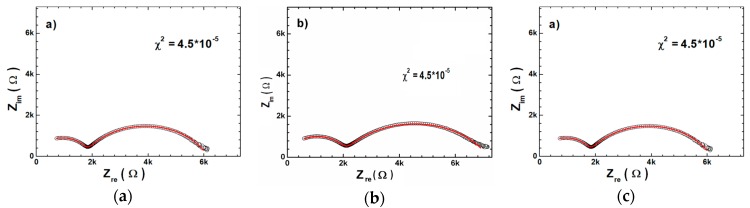
AC impedance spectrum in complex plane (open circles) and modeled impedance spectrum using calculated values of circuit elements (solid red line) for BCT20 (**a**) BCT25 (**b**) and BCT30 (**c**).

**Figure 10 materials-12-04036-f010:**
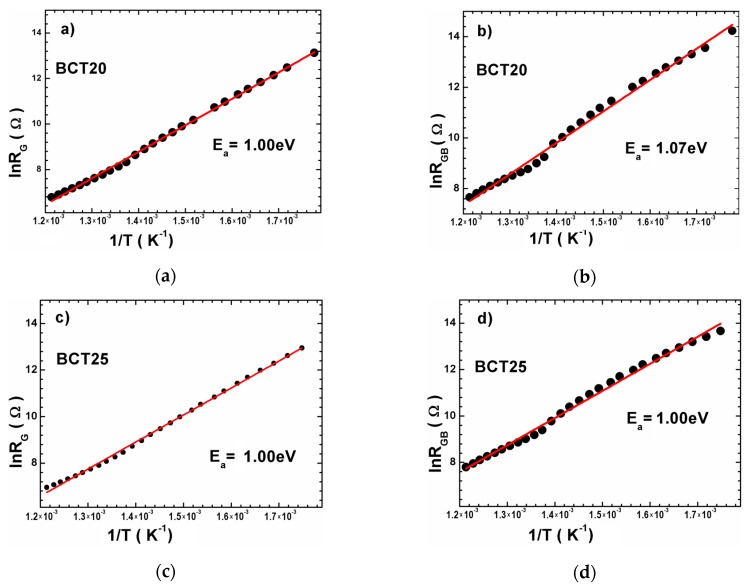
Plots for calculation of activation energies of grain (**a**,**c**,**e**) and grain boundary (**b**,**d**,**f**) for BCT20, BCT25 and BCT30 ceramics, respectively.

**Table 1 materials-12-04036-t001:** Relative density and grain size of the sintered BCT samples.

Sample	T_sd_ (°C)	T_md_ (°C)	Shrinkage (%)	Relative Density (%)
BCT20	1135	1358	14.12	0.98
BCT25	1127	1377	13.84	0.97
BCT30	1113	1379	13.55	0.95

**Table 2 materials-12-04036-t002:** EDS analysis of barium, calcium and titanium contents in the BCT ceramics.

Ceramics	Ba (%)	Ca (%)	Ti (%)	Ba + Ca/Ti
**BCT20**	39	10	51	0.96
**BCT25**	38	13	49	1.04
**BCT30**	37	15	48	1.08

**Table 3 materials-12-04036-t003:** The XRD related coefficients for obtained BCT samples.

Parameter	Unit	BCT20	BCT25	BCT30
Density calculated (XRD)	(g/cm^3^)	5.7612	5.7899	8.001
Tetragonality (c/a)	(−)	1.0106	1.0111	1.0118
Lattice parameter a	(Å)	3.9827(9)	3.9756(9)	3.977(1)
Lattice parameter b	(Å)	3.9827(9)	3.9756(9)	3.977(1)
Lattice parameter c	(Å)	4.025(1)	4.020(1)	4.024(2)
Unit cell volume V	(10^6^ pm^3^)	63.8	63.5	63.6
Space group	(−)	P4 mm	P4 mm	P4 mm

**Table 4 materials-12-04036-t004:** T_C_, ε′_max_, T_dev_, C, ΔT and γ obtained from measurements of ε’(T) at f = 10 kHz for BCT20, BCT25 and BCT30 ceramics.

Ceramics	T_C_ (K)	ε′_max_	T_dev_ (K)	C × 10^5^ (K)	ΔT (K)	T_0_ (K)	γ	T_C_ (K)
**BCT 20**	388	4936	424	0.95	36	379	1.32	388
**BCT 25**	390	4833	420	1.00	30	380	1.39	390
**BCT 30**	392	5299	419	0.96	27	384	1.58	392

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
