# Peer review of "Dielectric and Impedance Studies of (Ba,Ca)TiO3 Ceramics Obtained from Mechanically Synthesized Powders"

_materials, 2019, doi:10.3390/ma12244036_

Round 1
Reviewer 1 Report
This study on (Ba, Ca) TiO3 is clear and interesting.
However, I have some suggestions listed below.
English should be revised, there are some mistakes.
Line 20: Authors should write “… and consists in the introduction…”.
Line 22: Authors should write “… occupies the B site…”
Line 66: How were the sintering conditions determined? This should explained.
Lines 71,72, 247, 251: The style of the text is different from the rest of the article.
Line 247: Authors should write “… via mechanochemical…”
Author Response
Line 20: Authors should write “… and consists in the introduction…”.
The obtained results indicate, that the two mechanism of doping occurred. The first one is observed for the lower calcium concentration (below 0.3) and consists the introduction of calcium ion into the A site of perovskite structure.
Line 22: Authors should write “… occupies the B site…”
The second one is observed for the higher calcium concentration (equal 0.3). In this case the calcium ions partially occupies the B site in the perovskite structure. Both cases have different influence on the final properties of the ceramics because they induce different defects.
Line 66: How were the sintering conditions determined? This should explained.
Linear shrinkage rates facilitate the study of sintering kinetics involved in the calcium and barium titanate thickening process. Sintering studies (Ba, Ca) TiO3 were performed in a NETZSCH DIL 402 C dilatometer. Linear shrinkages for the green BCT samples after dilatometric experiments is clearly related to the compositions of investigated BCT.
Thus, increasing the amount of calcium in investigated materials results in a slight decrease in the initial compaction temperature, an increase in temperature where the maximum densification rate occurs, and marked reduction in overall shrinkage. Total shrinkage of samples, temperature of densification start Tsd and temperature of maximum densification Tmd for individual samples are summarized in Table 1, neither hardness nor fracture toughness were examined. A BCT20 sample with a near theoretical density (over 98%) was obtained. The BCT30 sample is more porous than others, probably due to the coexistence of two phases of barium titanate and calcium titanate with markedly different grain size as well as chemical and physical properties.
Lines 71,72, 247, 251: The style of the text is different from the rest of the article.
The obtained morphologies of the powders were examined using a scanning electron microscope with an ultra-high resolution JEOL JSM-7100F TTL LV with a cold field emission source. SEM powders were prepared by dissolving in ethanol in an ultrasonic bath.
Polycrystalline Ba1-xCaxTiO3 (BCT, x=0.2, 0.25, and 0.3) ceramics have been successfully obtained via mechanochemical synthesis. The microstructure of the resulting ceramics consists of very well developed grains with visible terraces, which indicates a tendency to spiral hexagonal growth. The increase of calcium content affects the final morphology which exhibits a significant increase of grain heterogeneity.
The results obtained suggest a change in the calcium substitution mechanism for BCT30 ceramics.
Line 247: Authors should write “… via mechanochemical…”
Polycrystalline Ba1-xCaxTiO3 (BCT, x=0.2, 0.25, and 0.3) ceramics have been successfully obtained via mechanochemical synthesis.
Reviewer 2 Report
This work presents results of an investigation of the formation and properties of (Ba,Ca)TiO3 ceramics. The powder was synthesized by the mechanochemical method. The concentration of Ca in the reaction mixtures was varied and the doping mechanism was studied. I think that the paper has a high potential to be accepted. I suggest minor revision.
The information on the raw materials (oxides) should be provided (particle size, purity, manufacturer).
How were the sintering conditions selected?
What is the relative density of the sintered ceramics? Why was the BCT30 sample more porous than the others? Are the sintered BCT materials strong enough mechanically?
Do you consider using sintering techniques other than conventional sintering? Spark Plasma Sintering or others? This could help in controlling grain growth and achieving better densification. Please comment on the choice of the sintering method.
Author Response
Author's Reply to the Review Report (Reviewer 2)
This work presents results of an investigation of the formation and properties of (Ba,Ca)TiO3 ceramics. The powder was synthesized by the mechanochemical method. The concentration of Ca in the reaction mixtures was varied and the doping mechanism was studied. I think that the paper has a high potential to be accepted. I suggest minor revision.
The information on the raw materials (oxides) should be provided (particle size, purity, manufacturer).
How were the sintering conditions selected?
What is the relative density of the sintered ceramics? Why was the BCT30 sample more porous than the others? Are the sintered BCT materials strong enough mechanically?
In the Materials and Methods section the missing data was included (yellow marked)
Materials and MethodsVarious compositions of BCT have been chosen for investigations: Ba0.8Ca0.2TiO3 - denoted as BCT20, Ba0.75Ca0.25TiO3 – BCT25 and Ba0.7Ca0.3TiO3 – BCT30.
The high-purity oxides BaO (99.9% pure, Sigma-Aldrich), CaO (99% pure, Avantor Performance Materials) and TiO2-anatase (99% pure, Sigma-Aldrich) were weighed out in the desired molar ratios. The constituent oxides were of standard powder particle average sizes in the range of 10-50 microns. All the powder samples were subsequently milled in the high-energy ball mill SPEX 8000 for 20h (according to [11]). Linear shrinkage rates facilitate the study of sintering kinetics involved in the calcium and barium titanate thickening process. Sintering studies (Ba, Ca) TiO3 were performed in a NETZSCH DIL 402 C dilatometer. Linear shrinkages for the green BCT samples after dilatometric experiments is clearly related to the compositions of investigated BCT.
Thus, increasing the amount of calcium in investigated materials results in a slight decrease in the initial compaction temperature, an increase in temperature where the maximum densification rate occurs, and marked reduction in overall shrinkage. Total shrinkage of samples, temperature of densification start Tsd and temperature of maximum densification Tmd for individual samples are summarized in Table 1, neither hardness nor fracture toughness were examined. A BCT20 sample with a near theoretical density (over 98%) was obtained. The BCT30 sample is more porous than others, probably due to the coexistence of two phases of barium titanate and calcium titanate with markedly different grain size as well as chemical and physical properties.
Table 1. Relative density and grain size of the sintered BCT samples.
|
Sample |
Tsd [°C] |
Tmd [°C] |
Shrinkage [%] |
Relative density [%] |
|
BCT20 |
1135 |
1358 |
14.12 |
0.98 |
|
BCT25 |
1127 |
1377 |
13.84 |
0.97 |
|
BCT30 |
1113 |
1379 |
13.55 |
0.95 |
Do you consider using sintering techniques other than conventional sintering? Spark Plasma Sintering or others? This could help in controlling grain growth and achieving better densification. Please comment on the choice of the sintering method.
In the last conclusions section the comment was included (yellow marked)
However, for the purpose of this experiment, the conventional mixed oxide ceramic (MOM) method was used as the first approach despite the fact that, for example, rapid sintering can significantly reduce processing time. As shown above, high temperatures and long reaction times in the MOM method give ceramics with very heterogeneous particle sizes. With the miniaturization of electronic devices, there is a great industrial need to control size effects, especially when approaching the nanosize scale. According to a report on a similar solid solution of Ba1-xSrxTiO3, the combination of mechanosynthesis and SPS offers compaction that took place in a few minutes, demonstrating that synergy occurs due to the combination of the SPS process and mechanosynthesized powders, which is why this method is planned in the future [25].